# Emerging and Biological Concepts in Pediatric High-Grade Gliomas

**DOI:** 10.3390/cells13171492

**Published:** 2024-09-05

**Authors:** Abigail Yoel, Shazia Adjumain, Yuqing Liang, Paul Daniel, Ron Firestein, Vanessa Tsui

**Affiliations:** 1Centre for Cancer Research, Hudson Institute of Medical Research, Monash University, Clayton, VIC 3168, Australia; abigail.yoel@hudson.org.au (A.Y.); ihara.adjumain@monash.edu (S.A.); yuqing.liang@monash.edu (Y.L.); paul.daniel@hudson.org.au (P.D.); ron.firestein@hudson.org.au (R.F.); 2Department of Molecular and Translational Science, Faculty of Medicine, Nursing and Health Sciences, Monash University, Clayton, VIC 3168, Australia

**Keywords:** pediatric, high-grade glioma, H3K27M, H3G34V/R, clinical trials, treatment, immunotherapy

## Abstract

Primary central nervous system tumors are the most frequent solid tumors in children, accounting for over 40% of all childhood brain tumor deaths, specifically high-grade gliomas. Compared with pediatric low-grade gliomas (pLGGs), pediatric high-grade gliomas (pHGGs) have an abysmal survival rate. The WHO CNS classification identifies four subtypes of pHGGs, including Grade 4 Diffuse midline glioma H3K27-altered, Grade 4 Diffuse hemispheric gliomas H3-G34-mutant, Grade 4 pediatric-type high-grade glioma H3-wildtype and IDH-wildtype, and infant-type hemispheric gliomas. In recent years, we have seen promising advancements in treatment strategies for pediatric high-grade gliomas, including immunotherapy, CAR-T cell therapy, and vaccine approaches, which are currently undergoing clinical trials. These therapies are underscored by the integration of molecular features that further stratify HGG subtypes. Herein, we will discuss the molecular features of pediatric high-grade gliomas and the evolving landscape for treating these challenging tumors.

## 1. Introduction

Primary central nervous system (brain and spinal cord) tumors are the most frequent solid tumors in children and the second most prevalent pediatric malignancy after leukemia [1]. Gliomas constitute 80% of all malignant primary CNS tumors, making them the leading cause of CNS tumor-related mortality, surpassing acute lymphoblastic leukemia [2,3]. Pediatric high-grade gliomas (pHGGs) mainly consist of anaplastic astrocytoma (WHO grade 3) and glioblastoma (WHO grade 4) [4], which are now recognized as IDH-mutant and IDH-wildtype tumors (4 and Louis et al., 2021 [5]). The global incidence rate of pHGGs is estimated to be between 1.1 and 1.78 cases per 100,000 children [6]. Although considered rare compared to adult HGGs (aHGGs), pHGGs account for over 40% of all childhood brain tumor deaths [7]. Reported overall survival for pHGGs ranges from 10 to 73 months [8]. pHGGs are most commonly found in adolescents aged 15–19 [9]. pHGGs have a five-year survival rate of less than 10%, even with aggressive treatment regimens [10,11,12]. Survival estimates for pHGGs vary based on their anatomical location, such as supratentorial, brainstem, or spinal cord. For tumors located in the supratentorial region, the 5-year overall survival rate is less than 20%, with most patients succumbing to the disease within 2 years of diagnosis [13]. As for tumors located in the brain stem, they also carry a dismal median survival of less than 1 year [14].

HGGs are most commonly diagnosed as primary tumors in children, although they can occasionally develop from LGGs [15]. Historically, pHGGs were classified together with adult gliomas due to the similarity in histological appearances and aggressive clinical behavior [16]. However, advances in genome and epigenome profiling technologies have shown that the origin and pathological features of pHGGs differ significantly from their adult counterparts [8,17,18]. These improvements have resulted in the reclassification of pHGG tumor subtypes by the WHO, incorporating molecular features as well as meaningful clinical data.

## 2. High-Grade Gliomas in a Nutshell

The latest classification of gliomas by the WHO integrates molecular diagnostics, histology, and immunohistochemistry data, providing descriptions for various glioma types. These include adult-type diffuse gliomas, pediatric-type diffuse low-grade gliomas, pediatric-type diffuse high-grade gliomas, circumscribed astrocytic gliomas, glioneuronal and neuronal tumors, and ependymal tumors [19,20,21,22]. Each type is further subdivided based on malignancy and its distinct molecular characteristics. The WHO CNS classification identifies four subtypes, including Grade 4 Diffuse midline glioma H3K27-altered, Grade 4 Diffuse hemispheric gliomas H3G34-mutant, Grade 4 pediatric-type high-grade gliomas H3-wildtype and IDH-wildtype, and infant-type hemispheric gliomas (Figure 1) [23]. 

### 2.1. Diffuse Midline Glioma (DMG) H3K27-Altered Subtype

In children, diffuse midline gliomas (DMGs) are primarily located in the pons and represent 15% of all pediatric brain tumors [7]. The 2021 WHO classification recognizes further subgroups of DMGs that are related to alterations in the H3K27M. These subtypes are characterized by their diffuse growth patterns and location along the midline (e.g., thalamus, brain stem, or spinal cord). Molecular features include the missense mutation of lysine to methionine (K27M) in the histone H3 genes H3F3A, HIST1H3B, HIST1H3C, or HIST2H3C [24,25]. The mutation affects the enzymatic activity of EZH2, a subunit of the Polycomb Repressive Complex 2 (PRC2) involved in gene silencing, leading to an extensive global loss of H3K27me3 methylation across the genome [26]. 

Additional changes, such as increased expression of the EZHIP protein, are also recognized as molecular markers for defining H3K27-altered tumors [27]. Emerging data indicate biological variations within K27M-mutated tumors where H3.3 mutations typically occur in midline structures and sometimes coincide with FGFR1 and/or NF1 mutations in certain thalamic gliomas. These usually occur in children aged 7–10 years and are linked with unfavorable outcomes [28]. In contrast, H3.1 mutations are predominantly seen in patients between the ages of 4 and 6 years. Children with H3.1 mutations exhibit distinct clinicopathologic and radiologic characteristics to the H3.3 mutations and they generally have slightly longer survivability, although prognosis is still dismal. These mutations often coincide with ACVR1 mutations [29,30].

### 2.2. Grade 4 Diffuse Hemispheric Glioma, H3G34-Mutant Subtype

Diffuse hemispheric glioma, H3G34-mutant, is a CNS WHO grade 4 astrocytoma that is diffusely infiltrative and arises in the cerebral hemisphere and is recognized as a distinct subtype from H3K27-altered tumors. Despite being classified as a glioma, its transcriptomic and epigenomic profiles suggest a neuronal origin [31]. This tumor primarily affects children and young adults, with a median age of 15 years [20]. It is mainly located in the temporal and parietal lobes [6]. This mutation is observed in around 16% of cortical pHGGs [32]. Patients with H3G34-mutant gliomas have only a slightly longer median survival (17.3 months) compared to patients with the K27M-altered glioma subtype (15 months) [33,34,35]. 

The driver alteration is a point mutation at codon 35 of the histone H3.3 gene H3-3A, which corresponds to glycine 34 in the mature H3.3 protein [36]. This mutation results in the substitution of glycine with either arginine (G34R) or, less frequently, valine (G34V). Consequently, it inhibits the activity of SETD2 methyltransferase and KDM2A lysine demethylase, resulting in widespread epigenetic remodeling characteristic of childhood gliomas [37,38,39].

### 2.3. Grade 4 Pediatric High-Grade Glioma H3-Wildtype and IDH-Wildtype

Diffuse pediatric-type high-grade glioma H3-wildtype and IDH-wildtype (pHGG H3/IDH WT) is a diverse entity currently characterized by an absence of oncohistone alterations (i.e., H3.3/H3.1/H3.2 pK27M) and methylation of three groups (i.e., pHGG RTK1, pHGG RTK2, pHGG MYCN) [40]. In this case, accurate diagnosis relies heavily on molecular characterization and the integration of both histopathological and molecular data [41]. pHGG H3/IDH-wildtype tumors are estimated to constitute approximately 40% of pHGGs. Among these, pHGG MYCN and pHGG RTK2 represent the largest and smallest subgroups, respectively [20].

The vast majority of pHGG H3/IDH-wildtype tumors occur in the supratentorial anatomic compartment [42]. The overall prognosis for pHGG H3/IDH-wildtype is poor. pHGG MYCN tumors are associated with the lowest survival rates, with pontine tumors within this subgroup behaving more aggressively compared to supratentorial counterparts. Median overall survival is 16.5 months for supratentorial HGG-MYCN and 1.5 months for pontine HGG-MYCN [40,43].

### 2.4. Infant-Type Hemispheric Gliomas

The essential diagnostic criteria for infant-type hemispheric glioma encompass a combination of clinicopathological and molecular characteristics [20]. Infant-type hemispheric glioma is commonly initiated by a fusion event involving a receptor tyrosine kinase (RTK) gene, such as NTRK1/2/3, ALK, ROS1, or MET [44,45]. The clinical presentation is typically acute, often occurring within the first year of life. Symptoms are nonspecific and can range from seizures to lethargy or irritability. Congenital cases are also reported, characterized by increased head circumference and bulging fontanelles, which are common clinical signs [46]. Historically, infant-type hemispheric gliomas have shown a poor response to standard chemotherapy and radiation treatments. Fusions with receptor tyrosine kinase genes (such as *ALK*, *ROS1*, *NTRK,* and MET) are highly prevalent and preclinical studies show promising results for kinase inhibitors for this glioma type [10,45,47]. The underlying cause of the poor overall survival, whether it stems from the common anatomical location of this disease or results from phenotypic convergence among various subgroups of pHGGs, remains a crucial question for future research investigation.

## 3. Clinical Diagnosis of pHGGs

pHGGs are known for their aggressive behavior and poor prognosis. Diagnosing these malignancies requires a comprehensive strategy that combines thorough clinical evaluation, advanced imaging methods, and detailed histopathological analysis. Symptomatically, children may present with signs of increased intracranial pressure including persistent headaches, behavioral changes, early morning nausea/vomiting, double vision (diplopia), and swelling of the optic nerve (papilledema). Additionally, patients may present with more specific symptoms related to tumor locations, such as muscle weakness, hemiplegia, dysmetria, and chorea [15]. While patients with HGGs may experience seizures, which commonly occur when the tumor invades the temporal lobe, it is not a typical presentation at diagnosis [48]. 

### 3.1. Imaging Techniques

Upon initial presentation, patients typically undergo a CT scan, although a surgical biopsy is most commonly performed to make a diagnosis. If a bleed or mass is detected, a contrast-enhanced MRI is then performed [49]. MRI is considered the gold standard for brain tumor imaging because of its high soft tissue contrast and ability to capture images in multiple planes [50]. In diagnosing pediatric brain tumors, MRI is utilized for pre-surgical planning, immediate postoperative imaging, intraoperative scans, follow-up imaging, creating individualized clinical plans, and planning radiation therapy [51]. One of the primary goals of MRI is to refine the differential diagnosis by evaluating the lesion’s characteristics across various imaging modalities, including T1, T2, T1 with contrast (T1 C+), apparent diffusion coefficient (ADC), diffusion-weighted imaging (DWI), susceptibility-weighted imaging (SWI), and fluid-attenuated inversion recovery (FLAIR), among others [52]. 

### 3.2. Histopathological and Immunohistochemistry Diagnosis 

Histopathological and immunohistochemistry (IHC) are critical components in the diagnosis and characterization of pHGGs [53]. Histopathologically, pHGGs exhibit distinct features such as high cellularity, marked nuclear atypia, frequent mitoses, necrosis, and microvascular proliferation, which are indicative of their aggressive nature. Examination of tissue samples obtained via biopsy or surgical resection under a microscope allows pathologists to determine the tumor grade and identify specific histological subtypes, which are essential for accurate diagnosis and treatment planning [54]. IHC further enhances the diagnostic process by using antibodies to detect specific antigens in the tumor tissue. This technique helps differentiate between tumor types and assess the expression of various molecular markers. Commonly used markers in pHGGs include GFAP (glial fibrillary acidic protein), which confirms glial origin; Ki-67, which indicates proliferative activity; and mutant proteins such as H3K27M, which is associated with a specific and aggressive subset of these tumors [55]. The combination of histopathological evaluation and IHC provides a comprehensive understanding of the tumor’s biological characteristics, aiding in prognostication and the development of targeted therapeutic strategies. 

### 3.3. Liquid Biopsy

A liquid biopsy offers an alternative means of gathering crucial tumor information from bodily fluids like blood (serum or plasma), cerebrospinal fluid (CSF), and urine, presenting a less invasive approach [56]. These fluids harbor biomaterials such as cell-free DNA, circulating tumor cells, circulating tumor DNA, fragmented peptides, and microRNAs, all of which are released from tumors [57]. Whole-genome sequencing of liquid biopsies is becoming increasingly popular for detecting H3 mutations in pHGGs due to the method’s precision in identifying single and multiple mutations in cancerous cells, along with the accessibility of tumor materials in liquid biopsies [58]. CSF stands out as a key source of biomarkers because of its higher concentration of circulating tumor DNA. However, obtaining CSF samples requires a lumbar puncture, an invasive procedure in which the risk–benefit should be carefully considered for each patient. This is particularly concerning since gliomas in the brain can elevate intracranial pressure, posing a risk of brain herniation due to the acute pressure created by CSF withdrawal [57]. Another possible way to access CSF is to approach using external ventricular drainage (EVD), which is found to be an overall safe and effective option in children [59]. 

## 4. The Molecular Landscape of pHGGs

### 4.1. Genetic Mutations in pHGGs

Mutations serve as a hallmark of cancer and are pivotal for comprehending the mechanisms underlying cancer development [60]. Hence, grasping the genomic mutations that trigger the dysregulation of cellular mechanisms and drive tumorigenesis constituted a fundamental principle of cancer research. The majority of pHGGs exhibit genetic complexity, which includes notable copy number alterations (CNAs), single nucleotide variants (SNVs), and structural variants [61,62,63]. Several molecular pathways, characterized by specific genetic mutations, are believed to play key roles in tumorigenesis and will be explored in the subsequent sections.

#### 4.1.1. Receptor Tyrosine Kinase (RTK)

RTKs are transmembrane proteins with intrinsic enzymatic activity, crucial for signaling pathways involved in cell proliferation, differentiation, and survival. The RTK family includes PDGFR, epidermal growth factor receptors (EGFRs), and fibroblast growth factor receptors (FGFRs), among others [64]. In DMGs, over 60% exhibit various amplifications and mutations affecting components within the RTK-RAS-P13K pathway. These genetic alterations frequently coincide with the H3K27-altered subgroup [65]. Recently, the WHO has categorized EGFR-mutant gliomas as a separate subtype of H3K27-altered gliomas, characterized by primary abnormalities occurring within the EGFR oncogene on chromosome band 7 [6]. Targeting this specific EGFR alteration holds promise as a potential treatment that may benefit these patients, but further research is required [31]. EGFR mutations are more prevalent in adult populations, affecting approximately 90% of aHGGs. In contrast, EGFR mutations are less common in pediatric populations, with gene amplification and EGFRvIII overexpression detected in only a small percentage (4%) of pHGGs [66,67].

#### 4.1.2. Tumor Protein p53 (TP53)

The tumor suppressor gene TP53 encodes the p53 protein, serving as a transcription factor crucial for tumor suppression [68]. Various stress signals, including DNA damage, hypoxia, and chemotherapy, activate the p53 pathway, which in turn triggers different cellular responses such as cell cycle arrest, apoptosis, differentiation, DNA repair, and autophagy through intricate networks [69,70]. The most frequent mutations affecting the p53 pathway include missense mutations in TP53, deletions of CDKN2A/ARF, and/or amplification of MDM2 and MDM4. These alterations result in reduced tumor suppressor activity [71]. TP53 mutations have been observed in up to 80% of DMG samples, often alongside the H3K27 alterations. However, they are also found in H3-wildtype tumors [72]. In a retrospective analysis, it was observed that DMGs with mutations in both H3K27-altered and TP53 exhibit increased resistance to radiation therapy, enhanced tumor aggressiveness, and worse overall survival compared to patients without these mutations or with only one mutation present [73]. Indeed, while the H3K27M mutation may serve as the primary oncogenic driver, the co-occurrence of both mutations and the resulting worse clinical outcomes suggest that the multifactorial molecular alterations in DMGs contribute to its fatal nature [65].

#### 4.1.3. Activin A Receptor, Type 1 (ACVR1)

ACVR1, a member of the TGF-beta signaling family, is a bone morphogenic protein (BMP) receptor that binds to various ligands. ACVR1 activation leads to the phosphorylation and activation of growth-promoting genes through SMAD transcription factors [74,75]. Certain mutations within ACVR1 are part of the molecular profile of DMGs. Somatic mutations such as R206H, R258G, G328E/V/W, and G356D within ACVR1 have been identified in up to 20% of DMG cases, according to some retrospective analyses [30,76]. Under normal, unmutated conditions, ACVR1 aids in myelination within the CNS [77]. When a mutation occurs in ACVR1, it encodes a serine/threonine kinase (ALK2) receptor with heightened sensitivity to the ligand activin A. This results in dysregulation of the BMP/SMAD pathway and increased tumor proliferation. The mutation is also associated with an earlier age of tumor development, with a median age of diagnosis of 5 years, and a slightly improved overall survival of 15 months [78].

#### 4.1.4. ATRX

Inactivating mutations in the chromatin remodeler ATRX are identified in hemispheric pHGG, closely associated with H3F3A-G34R/V mutations, and are found in both the IDH and histone wildtype epigenetic subtypes [79]. ATRX inactivation is described as essential for triggering the alternative lengthening of telomeres (ALT) mechanism. This mechanism enables pHGG cells to extend their telomeres without needing telomerase reverse transcriptase (TERT) expression, thereby preventing cell death from progressive telomere shortening. This allows cancer cells to divide indefinitely, facilitating cancer progression [80]. ATRX is an SNF2 helicase/ATPase that collaborates with DAXX to create a H3.3 chaperone complex. This indicates that H3.3 and its ATRX chaperone complex play a fundamental role in the development of pediatric gliomas (Voon & Wong, 2023 [81]). A study showed that approximately 17% of all pHGGs have inactivating mutations in the ATRX gene [6]. Among these ATRX-mutated HGGs, 33% also carry H3.3 G34R/V mutations and 50% overlap with H3.3 K27M mutations. Notably, there is no overlap between ATRX and H3.1 K27M mutations, which are instead linked to ACVR1 mutations. These findings strongly suggest that histone H3.3 has a significant oncogenic role in pHGGs [81]. 

#### 4.1.5. BRAF V600E Mutation

The V-RAF murine viral oncogene homolog B1 (BRAF) is a member of the RAF1 serine/threonine protein kinase family and acts as an oncogene in various malignancies, including primary tumors of the central nervous system (CNS). Under normal physiological conditions, BRAF is activated by RAS (Rat Sarcoma virus) proteins, which are small GTPase proteins [82]. Once activated, BRAF forms homo- or hetero-dimers, which then activate the mitogen-activated protein kinases MEK1 and MEK2 through phosphorylation. Notably, MEK1 and MEK2 are encoded by the genes mitogen-activated protein kinase 1 (MAPK1) and mitogen-activated protein kinase 2 (MAPK2), respectively [83]. The most common mutation in BRAF is c.1799T>A, resulting in the substitution of valine with glutamic acid at position 600 (V600E). This mutation is frequently detected in gliomas [84]. 

In pediatric cohorts, BRAF alterations are primarily observed in low-grade gliomas, including pilocytic astrocytoma and glial-neuronal tumors [85]. Data from large studies indicate that BRAF V600E-mutated HGGs constitute only 6–15% of all pHGGs. The prognosis for BRAF V600E mutations in pHGGs seems more favorable compared to other molecular groups, with a 2-year survival rate of 67% [6]. However, the prognosis also varies depending on histological subtypes and associated molecular alterations [86]. With the advancement in research, there are now several targeted therapies available for managing these aggressive brain tumors. BRAF inhibitors (e.g., Vemurafenib and Dabrafenib) and MEK inhibitors (e.g., Trametinib and Selumetinib) are some of the few targeted therapies for treating pHGGs with BRAF V600E mutations [86,87]. 

#### 4.1.6. Neurofibromatosis Type 1 (NF-1)

Brain tumors associated with neurofibromatosis type 1 (NF1) are typically found in LGGs [88]. Mutations affecting neurofibromin, a protein involved in regulating cell growth and the Ras proto-oncogene, can lead to unchecked cell proliferation and the development of either LGGs or HGGs. In children with HGG, the occurrence of NF1 is relatively uncommon, ranging from 0.28% to 5% [89]. Individuals with NF1 have a significantly higher risk of developing low-grade gliomas compared to high-grade gliomas. However, their risk of developing high-grade glioma is increased by 50-fold compared to the general population [90]. Molecular analyses of NF1-associated pHGGs have identified mutational and genomic alterations akin to those in their sporadic counterparts. These include mutations in the ATRX, TP53, and CDKN2A genes, as well as in genes involved in the phosphoinositol-3 kinase (PI3K) pathway. Notably, NF1-associated high-grade gliomas do not exhibit the IDH and histone H3 mutations frequently found in sporadic malignant gliomas [91,92]. 

#### 4.1.7. Neurotrophic Tyrosine Receptor Kinase (NTRK) Fusion

Recent studies have demonstrated recurrent fusion of the neurotrophic tyrosine receptor kinase (NTRK) gene in 10% of high-grade gliomas outside the brainstem in very young children, indicating that NTRK fusion genes have an oncogenic impact [45]. A study conducted by Garcia et al. (2022) [93] found that the NTRK genes (i.e., NTRK1, NTRK2, and NTRK3) are implicated in infant-type hemispheric gliomas, which typically exhibit high-grade histology. A recent study has indicated that most gliomas fused with NTRK genes are located in the hemispheres and are more prevalent among high-grade gliomas outside the brainstem in patients younger than 3 years old [94,95]. These tumors historically have been associated with high mortality and recurrence due to their aggressive nature and high-grade histology. However, the prognosis for NTRK-fused gliomas may be changing following the recent FDA approval of selective pan-TRK inhibitors such as larotrectinib, entrectinib, and repotrectinib [96,97]. 

## 5. The Tumor Microenvironment in pHGGs

The tumor microenvironment (TME) comprises non-cancerous cells within and around the tumor, including immune cells, endothelial cells, microglia, astrocytes, and neurons. pHGGs exhibit an immunologically inactive tumor microenvironment, characterized by low T cell infiltration and limited immune surveillance [98]. Because of its varied components and ever-changing nature, the TME is crucial in determining cancer cell survival and their response to treatment [99]. 

The TME contributes to additional heterogeneity among these tumors. Another layer of complexity lies in the maturation of the immune system during childhood, so caution is necessary before applying findings about the TME from adult brain tumors to pediatric types. Analysis reveals immune infiltration in pLGGs and pHGGs, indicated by the presence of CD163+ macrophages and CD8+ T cells. However, for DMGs, there is no increase in any immune cells compared to normal tissue controls [100]. They are also found to be lacking in cytokines and chemokines (i.e., IL6, IL1A, and CCL3, etc.) necessary to recruit immune cells [100,101]. In children, glioma cells also establish an immunosuppressive environment through various mechanisms, including the release of soluble factors and induction of hypoxia. For instance, TGF-β inhibits T cell proliferation and IL-2 production, activates regulatory T cells (Tregs), and downregulates NKGD2 [102,103].

Research has demonstrated that the immune landscapes across all pediatric brain tumors do not correlate with tumor grade, mutational burden, mesenchymal/epithelial phenotype, or patient prognosis [104]. IHC data revealed the opposite trend, with more CD45+, CD8+, and PD1+ cells present in low-grade gliomas compared to higher-grade tumors. T cell receptor sequencing also indicated a trend toward greater T cell infiltrate and clonality in low-grade tumors. Conversely, flow cytometry data showed a trend towards increased B cell numbers and B cell activation in high-grade gliomas [103]. Furthermore, flow cytometric analysis of adult gliomas and brain metastases revealed that IDH mutation status and tumor origin significantly influence the TME. Gliomas, in comparison to brain metastases, showed lower lymphocyte counts and a higher presence of microglia- and monocyte-derived infiltrating macrophages. Additionally, glioblastomas with wildtype IDH status demonstrated more lymphocyte and less macrophage infiltration compared to lower-grade IDH-mutant gliomas [105]. This correlation between tumor grading and lymphocyte infiltration has not been observed in pediatric subtypes [106]. 

## 6. Current Standard of Care for pHGG Patients

The aggressive nature of HGGs has prompted the adoption of multi-modal treatment strategies. These typically involve surgical debulking followed by a combination of radiation therapy and/or chemotherapy, which is considered the standard of care.

### 6.1. Surgery

Surgical resection serves as the initial step in the treatment of HGGs [107]. The primary objectives of surgical resection in the treatment of HGGs are to obtain tissue for pathological diagnosis, alleviate intracranial pressure, and cytoreduce tumors [108]. However, complete tumor resection is often challenging due to the diffuse infiltrative nature of the tumors and the high risk of causing permanent neurological deficits [109]. In cases where complete tumor resection is not feasible due to the tumor’s diffuse infiltrative nature or its location in the eloquent cortex, stereotactic needle biopsy may be the only surgical option, especially for deep lesions [110]. However, for more superficial lesions or those outside eloquent areas, several studies have demonstrated that gross total resection (GTR) is associated with prolonged survival compared to subtotal resection (STR) or biopsy [111,112]. For hemispheric tumors, GTR has shown significant benefits in terms of prolonger survival (HR, 0.29; 95% CI, 0.15–0.54; *p* < 0.001), as well as for infratentorial tumors (HR, 0.44; 95% CI, 0.24–0.83; *p* = 0.01). However, this survival advantage is not as pronounced for midline tumors (HR, 0.63; 95% CI, 0.34–1.19; *p* = 0.16) [113]. While there has been improvement in surgical mortality rates for HGGs, a significant issue remains with morbidity [114]. Post-surgical complications for HGGs can include stroke, infections, brain edema, and neurological dysfunctions [115]. The decision to attempt surgical resection is also influenced by factors such as the patient’s clinical condition, age, associated hydrocephalus, and the surgeon’s assessment of the risk of neurological sequelae. These considerations highlight that surgery alone is not sufficient as a form of treatment for pHGGs [116,117].

### 6.2. Radiotherapy (RT)

Radiation therapy involves the administration of high-energy radiation to target and kill tumor cells, and it is the standard of care following surgical resection [118]. Radiation therapy plays a crucial role in improving the survival of children with HGGs. Due to the infiltrative nature of HGG lesions, even after complete tumor resection, microscopic tumor cells may still be present. Therefore, radiation therapy is administered to prolong patient survival by targeting and eliminating these residual tumor cells [119]. The utilization of radiation therapy is also contingent on factors such as the histologic diagnosis of the tumor, the availability of effective chemotherapeutic alternatives, and the age of the child. In pediatric patients, a radiation dose of 54 Gy is typically administered in 1.8 Gy daily fractions over a period of 6 weeks. However, radiation therapy is often avoided in children under 3 years old due to the risk of severe neurocognitive sequelae [120]. Research indicates that, compared to their adult counterparts, pediatric patients have a higher 6-month survival rate following radiation therapy—79% versus 47% in adults. This suggests that radiotherapy should be considered, when feasible, for all childhood gliomas. However, the long-term side effects of radiotherapy on the developing brain must be considered for pediatric patients. These include neurocognitive deficits, vasculopathy, endocrine dysfunction, growth defects, and secondary malignancies [121].

### 6.3. Chemotherapy

Chemotherapy involves administering chemical compounds to kill cancer cells. It can be given systemically or locally and may be used after surgery, before surgery, or as adjuvant therapy post-surgery [122,123,124]. Treating HGGs with drugs has been challenging because the drugs need to be both potent and cytotoxic while also being able to penetrate the blood–brain barrier and overcome mechanisms of drug resistance [125,126]. The initial major clinical trials that demonstrated the effectiveness of Temozolomide (TMZ) in treating HGG were conducted in adults, including those with relapsed disease, and later, in newly diagnosed patients [127,128]. These studies established the use of postoperative radiotherapy (RT) with TMZ followed by maintenance of TMZ as the standard treatment for adults with newly diagnosed glioblastoma. For pediatric populations, the maximum tolerated dose and toxicity profile or oral TMZ have been defined through two phase I pediatric studies [129]. Although it has been identified that there are no significant improvements to survival, TMZ has been adopted as the standard of care with newly diagnosed HGG (ACNS0126). This decision was based on its less toxic profile, tolerability, and the lack of alternative regimens showing substantial improvements in survival rates, and this standard has largely remained unchanged today [130]. Following up on the previous study, the ACNS0423 trial evaluated concomitant TMZ followed by adjuvant TMZ and lomustine and found moderately improved event-free survival (EFS) and overall survival compared to the historical cohort from the ACNS0126 trial. However, this regimen resulted in a substantial increase in hematological toxicities compared to TMZ alone, indicating that the role of adjuvant chemotherapy still needs further investigation [131,132]. 

Although studies showed that TMZ is not particularly effective in pHGG tumors over conventional treatment, the methylation site in MGMT has been found to be a predictor of TMZ sensitivity [133]. Hypermethylation of the MGMT gene promoter results in the silencing of gene expression, which leads to decreased levels of the MGMT enzyme, enhancing the cytotoxic effects of TMZ [134,135]. Within the population of pHGG tumors that are hypermethylated at MGMT, there seems to be greater TMZ activity. However, it is still unknown how this methylation site plays a role in pHGGs; therefore, there is an urgent imperative to develop targeted therapies tailored for pHGG patients to improve treatment outcomes and survival rates.

## 7. Emerging Therapies for pHGGs

Recent advancements therapies integrating the molecular profile of pHGGs has enabled the rapid translation of treatments in clinical trials (Table 1). 

### 7.1. Immunotherapy

Treating brain tumors has been particularly difficult due to the unique biological characteristics of these cancers that often hinder progress. Key challenges include the tumor’s location behind the blood–brain barrier (BBB) and the patient’s age [141]. Certain brain tumors, especially those in children, can be cured with aggressive surgery, radiotherapy, and chemotherapy. However, these treatments often come at a significant cost, particularly for young children, who may suffer lifelong neurocognitive and endocrine side effects [142]. Immunotherapy has emerged as a novel paradigm in cancer treatment over recent years. The effectiveness of immunomodulating strategies is thought to depend on the presence of cytotoxic immune cells in the tumor or peripheral blood, which could migrate to and eradicate the tumor. The process of tumor development leads to the production of tumor-specific neo-antigens. Despite this, cancer cells often evade detection. They develop various mechanisms to resist immune surveillance, including local immune evasion, induction of immune tolerance, and systemic disruption of T cell signaling [143]. Furthermore, due to the heterogeneous nature of tumor cells, the selection pressure from immune recognition results in an immune editing process, promoting the survival and growth of more resistant tumor cells [144]. 

Indeed, the immune system can detect and combat tumor cells, but this capability eventually fails, as evidenced by characteristics of the chronic immune response [145]. This failure is marked by T cell exhaustion, mediated by the expression of inhibitory receptors such as programmed cell death protein 1 (PD-1), T cell immunoglobulin and mucin domain-containing protein 3 (TIM-3), and lymphocyte activation gene 3 (LAG-3) [146]. In this context, tumors can create a microenvironment that promotes immune inhibition, preventing direct stimulation of the antitumor immune response. These immune checkpoints, or co-inhibitory pathways, are powerful tools tumors use to evade immune attacks, making them crucial targets for immunotherapies [147]. However, immune checkpoints are not the sole targets for intervention, and much of the current research on immunotherapy is centered around adults, often overlooking the fundamental distinction between pediatric and adult immune responses [148]. Indeed, while immunotherapies like CAR-T cell therapy and checkpoint inhibitors have shown efficacy in treating nervous system tumors, pHGGs remain largely refractory to treatment. This underscores the necessity for delving into the immune milieu of pHGGs to fully elucidate the potential of immunotherapy in their management. 

#### Checkpoint Blockade Inhibitors

Immune checkpoints are integral to the immune system’s regulation, ensuring that immune responses remain controlled to avoid damaging healthy tissues. Numerous other checkpoint blockades have demonstrated promise and improved patient survival across a variety of cancers [149]. However, as previously discussed, pHGGs, particularly DMG, have limited immune infiltration. Therefore, it is not surprising that these therapies have shown no survival benefit for pHGG patients, except for those with hypermutant tumors [150]. A deeper understanding of the tumor microenvironment in pHGGs may be crucial for elucidating the exact mechanisms behind the failure of current treatments. Currently, several ongoing phase 1 and phase 3 clinical trials are investigating the use of anti-PD1 and anti-CTLA4 therapies in both newly diagnosed and refractory pHGGs. These trials will likely provide valuable insights into the disease and potentially lead to improved patient outcomes [148,151].

### 7.2. Chimeric Antigen Receptor T Cell Therapy

Chimeric antigen receptors (CAR) are artificially engineered receptors that empower lymphocytes, typically T cells, to identify and eliminate cells expressing a particular target antigen. CAR-T therapy’s effectiveness against B cell lymphoblastic leukemia led to its approval by the FDA in 2017 [152,153]. However, CAR-T therapy’s efficacy against solid tumors, including HGG, is generally limited, and it can sometimes lead to life-threatening toxicities [154]. 

Although pHGGs have a low mutational burden and therefore fewer tumor-specific antigens available for targeting, the distinct expression of mutated histone proteins in H3K27M and H3G34R make them promising targets for CAR-T therapies [148]. To enhance therapeutic response, ongoing phase 1 clinical trials are exploring HER2, EGFR, and IL13RA2 CAR-T cells for both adult and pediatric HGGs. Leveraging the unique biological characteristics of pHGGs could optimize these efforts [155]. 

An encouraging avenue for CAR-T therapy in pHGGs involves B7-H3 CAR-T, targeting B7-H3, an immune checkpoint molecule with high expression in tumor tissue and low presence in normal tissue. Crucially, B7-H3 is highly expressed in pediatric solid tumors like HGGs and DMGs [156,157]. When B7-H3 CAR-T cells are co-cultured with patient-derived DMG lines, they exhibit high levels of interferon-gamma, tissue necrosis factor-alpha, and interleukin 2 production, indicating their cytotoxic potential [156]. Another promising avenue is CAR-T cell targeting of disialoganglioside GD2, as it has been demonstrated to be highly expressed on the surface of all H3K27M DMG cultures examined [158]. Additionally, GD2 levels in normal tissue are very low, while it is extensively expressed in multiple tumor types [159]. In vivo studies have shown that GD2-specific CAR-T cells exhibit specific cytotoxicity towards H3K27M DMG tumor cells but not H3WT DMG tumor cells. This cytotoxicity is driven by the production of interferon-gamma and interleukin 2 [158]. In the mouse xenograft study, significant tumor clearance was achieved; however, a small subset of GD2-negative tumor cells survived, underscoring the importance of combination therapies. These initial studies underscore the potential of CAR-T therapies in targeting pHGGs. Nonetheless, several issues need addressing, particularly treatment-induced edema, which can be fatal in tumors located near the brainstem in DMG cases.

### 7.3. Cancer Vaccines

Advances in tumor immunology, alongside technical improvements in vaccine development, have renewed interest in using vaccination as a cancer therapy [160,161]. Cancer vaccines aim to stimulate a patient’s adaptive immune system by exposing it to a high concentration of tumor antigens. Identifying the appropriate molecular targets is crucial for designing an effective and specific cancer vaccine. Once target antigens are selected, they are administered alongside immune adjuvants to effectively activate the host’s antigen-presenting cells (APCs). These APCs must then be able to induce sustained responses from CD4+ and CD8+ T lymphocytes [162]. Since tumor cells share much greater similarity with healthy tissue than with infectious microorganisms, the key to constructing successful cancer vaccines is to protect healthy tissue while inducing targeted immunity against tumor antigens that are preferentially expressed by tumor cells [163,164].

## 8. New Therapeutic Strategies for Treating pHGGs

### 8.1. Precision Medicine and Targeted Therapy

Despite a significant number of clinical trials for pHGGs, there has been little change in therapies for these patients over the past 50 years and patient outcomes remain abysmal. This can partly be attributed to poor drug penetration through the blood–brain barrier, ineffective drug perfusion, and intra- and inter-heterogeneity of the tumor. As pHGGs represent a relatively small proportion of cancers and are classified as “rare”, the limited sample sizes pose a challenge to identifying novel therapies.

The current treatment approaches for pHGG are considered outdated, offering limited survival benefits for patients. Immunotherapy has shown great potential as a therapeutic strategy, but personalized precision medicine is now becoming the focal point of treatment strategies [165]. This approach aims to target key oncogenic genes for disruption, tailoring treatment to the individual patient. Furthermore, precision medicine is actively identifying new targets for targeted therapy by conducting genetic profiling across various tumor types [166]. The objective is to develop personalized treatments tailored to individuals’ tumor characteristics. This approach aims to provide more precise health trajectories and enable the detection of disease progression. 

### 8.2. RTK Inhibitors

Alterations in RTKs are a hallmark of HGG tumors, making drugs targeting these alterations a focal point in HGG research. Numerous clinical trials have assessed various RTK-targeted therapies [167]. Examples include EGFR inhibitors (e.g., Gefitinib, Erlotinib, Afatinib, and AZD9291), the EGFRvIII peptide vaccine Rindopepimut, EGFR monoclonal antibodies, and PDGFR inhibitors (including Imatinib mesylate, Dasatinib, and Tandutinib), either as monotherapies or in combination with other agents like Temozolomide, Bevacizumab, or radiation therapy [168]. Regrettably, many of these therapies have failed to yield long-lasting responses in clinical trials. This is because several of these genes can activate redundant signaling pathways, rendering drugs directed toward single targets ineffective [169]. Despite a few anecdotal responses, the overall evidence underscores the lack of efficacy of these therapeutic agents.

### 8.3. Epigenetic Therapies

Given that epigenetic mutations like K27M and G34R/V are prevalent in HGGs, treatments targeting them are being developed and have demonstrated promising results in pre-clinical studies [170]. For instance, the use of GSK-J4, an inhibitor of the histone H3K27 demethylase JMJD3, has led to significant reductions in tumor growth and prolonged survival in vivo in K27M glioma xenograft models [171]. Numerous Phase 1 clinical trials have been conducted on adult HGG patients using HDAC inhibitors like panobinostat, bevacizumab, and romidepsin [172]. For instance, combining Panobinostat with stereotactic re-irradiation for treating recurrent HGG resulted in a progression-free survival (PFS) rate at 6 months of 83% in Phase I, with Phase II testing currently underway [173,174]. Moreover, clinical trials are ongoing for pediatric HGGs, investigating the efficacy of HDAC inhibitors such as panobinostat and vorinostat [172]. The combination of vironostat, bevacizumab, and temozolomide for treating HGGs showed a 6-month PFS rate of 52.4% in Phase I/II clinical trials, which appears promising [175]. Mutant IDH tumors exhibit a deficiency in the homologous recombination system for repairing double-stranded DNA breaks [176,177]. Poly ADP-ribose polymerase 1 (PARP1) inhibitors were consequently introduced to counteract the repair of single-strand DNA breaks and modulate chromatin remodeling through histone modifications. PARP is the cellular protein responsible for repairing damaged DNA [178]. Indeed, alongside IDH1 inhibitors like ivosidenib and enasidenib, PARP inhibitors such as olaparib and veliparib are currently undergoing testing, either as monotherapies or in combination with temozolomide, in children with newly diagnosed and recurrent IDH1/2 mutant HGG [179]. 

### 8.4. Functional Drug and Genetic Screens

Currently, there are variable responses to drug treatments among HGG patients, largely due to the intricate tumor heterogeneity and the acquisition of genetic mutations during treatment, leading to drug resistance [117]. Hence, there is an urgent need for tools that assess cell–drug responses to develop effective treatments aimed at improving patient survival. High-throughput screens are crucial in this regard, as they test various concentrations of each drug on glioma cell lines, helping identify potential drugs capable of inhibiting tumor growth [180]. Currently, numerous commercially available drug libraries are accessible for testing on malignant tumors, facilitating the discovery of novel therapeutics capable of inhibiting tumorigenesis [181]. These include drug libraries such as those from the U.S. Food and Drug Administration (FDA), Cambridge, and kinase inhibitor libraries (e.g., Selleckchem, Houston, TX), which comprise drugs at different stages of clinical investigation for various cancers, targeting a wide array of molecular pathways. Translating drug data into clinical practice necessitates the identification of biomarkers and potential mechanisms. Therefore, alternative approaches like functional genomics and sequencing technologies may offer additional evidence to expedite translation efforts. 

Pediatric cancers are often described as having “quiescent” genomes because they exhibit a lower mutational burden compared to their adult counterparts [182]. However, research by multiple groups shows that mutational burden does not correlate with fewer genetic dependencies [18,183]. A first-generation pediatric dependency map showed that it was indeed possible to identify genetic dependencies across pediatric cancers, and there remains a broader range of therapeutic targets to be discovered [183]. Similarly, the Childhood Cancer Model Atlas uncovered specific genetic vulnerabilities that were therapeutically actionable by readily available inhibitors [18].

## 9. Conclusions and Future Directions

HGGs in pediatric patients remain among the most challenging cancers to treat, with poor prognosis despite advances in treatment modalities. Future research directions are recommended to focus on several critical areas to improve treatment outcomes and patient quality of life. Comprehensive genomic and epigenetic profiling should continue to identify novel mutations and pathways, enhancing our understanding of the molecular landscape of pHGGs. This could lead to the development of new targeted therapies, which are crucial given the heterogeneous nature of these tumors. Investigating combination therapies that synergize targeted treatments with existing modalities like radiation and chemotherapy is another promising avenue. Immunotherapy, which has shown success in adult gliomas, should be explored for pHGGs, particularly checkpoint inhibitors, cancer vaccines, and adoptive cell transfer methods. Personalized precision medicine, guided by biomarker discovery and patient-derived models, holds significant promise in tailoring treatments to individual tumor profiles, potentially improving therapeutic efficacy and minimizing side effects. 

Clinical trial groups such as the Pacific Pediatric Neuro-Oncology Consortium (PNOC) and the Collaborative Network for Neuro-oncology Clinical Trials (CONNECT) aim to develop trials driven by relevant biological research for rapid clinical translation and expand access to innovative therapies for children with brain tumors. Expanding innovative clinical trials and fostering international collaboration through global research networks will accelerate the development and testing of new therapies. Additionally, focusing on supportive care, including comprehensive palliative care and neurocognitive rehabilitation, is crucial to improving the quality of life for pHGG patients and survivors. By addressing these areas, future research aims to significantly improve survival rates and quality of life for children affected by these aggressive tumors.

## Figures and Tables

**Figure 1 cells-13-01492-f001:**
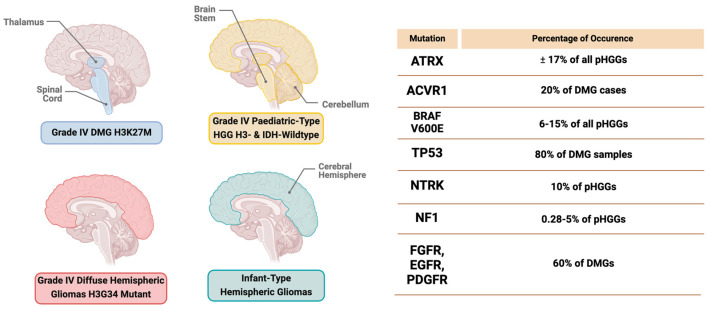
Locations of grade 4 high−grade glioma subtypes in the brain with mutation frequency.

**Table 1 cells-13-01492-t001:** Emerging clinical therapies/treatments for managing pHGGs.

Inhibitors, Immunotherapy or Others	NCT #	Phase of Clinical Trial	Target Gene/s	Cohort
NK cells injection	NCT04254419	1	MGMT, BRAFV600E, ACVR1, ATRx, TP53, H3G34, H3.3/H3.1K27, IDH1, CDKN2A, PDGFR	Recurrent HGG
Intratumoral injection of G207 + RT [136]	NCT04482933	2	-	Neoplasms, HGG, astrocytoma
Lutathera (177Lu-DOTATATE)	NCT05278208	1 and 2	Type-2A somatostatin receptors (SST2A)	HGG, medulloblastoma
iC9-GD2-CAR-T cells	NCT05298995	1	GD2	Medulloblastoma, HGG, DMG
Loc3CART: Locoregional Delivery of B7-H3-CAR T cells	NCT05835687	1	-	HGG (B7-H3-positive)
CLR 131	NCT03478462	1	Breakdown of dsDNA	DMG, HGG
Berubicin	NCT05082493	1	Topoisomerase II (Topo II)	Progressive, refractory, or recurrent HGG
NEO100	NCT06357377	1	*Rad*/*Raf* pathway	HGG, DMG
AsiDNA (etidaligide) + RT	NCT05394558	1 and 2	DNA-PK	HGG, DMG
Larotrectinib [137]	NCT04655404	1	TRK	HGG with NTRK fusion, DMG
LAM561 acid	NCT04299191	1 and 2	MAPK, CDK, PI3K inhibitor	HGG
Selinezor + RT	NCT05099003	1 and 2	CRM1	HGG (H3K27M-mutant or H3K27-WT), DMG
Fimepinostat	NCT03893487	1	PI3K, HDAC	DIPG, HGG, medulloblastoma
DC vaccine + TMZ [134,138]	NCT04911621	1 and 2	Anti-tumor defense mechanisms	HGG, DIPG
Trametinib + Everolimus	NCT04485559	1	MEK, mTOR	Recurrent HGG, grade 2 glioma, LGG
Novel peptide vaccine (PEP-CMV) + TMZ + Tetanus–Diphtheria vaccine	NCT05096481	2	-	Medulloblastoma, HGG, DIPG
Lorlatinib + BABYPOG/HIT-SKK	NCT06333899	1	ALK, TRK receptors	HGG with ROS1 or ALK fusion
Abemaciclib + TMZ + RT	NCT06413706	2	CKD4/6, *Rb* phosphorylation	HGG
C7R-GD2 CAR-T cells + cyclophosphamide and fludarabine [139]	NCT04099797	2	GD2	GAIL-B in DMG, HGG, DIPG, medulloblastoma
Ribociclib + Everolimus	NCT05843253	2	CDK4/6, growth-driven transduction signals in T cell response	HGG and DIPG with PI3K/mTOR mutations
Dabrafenib, Trametinib and Hydroxychloroquine	NCT04201457	1 and 2	BRAFV600E, MEK	HGG with BRAFV600E, BRAF fusion/duplication positive, or NF1-associated mutations
Olutasidenib + TMZ	NCT06161974	2	IDH1	HGG with IDH1 mutations
Panobinostat + Everolimus	NCT03632317	2	Histone deacetylase	H3.1 or H3.3 K27M DIPG
Neo-antigen HSP vaccine (rHSC-DIPGVax) + Balstilimab/Zalifrelimab [140]	NCT04943848	1	16 peptides on DIPG and DMG	DIPG, DMG
Dabrafenib + Trametinib	NCT03975829	4	BRAF, MEK1/2	HGG
Topotecan + TMZ + Ribociclib	NCT05429502	1 and 2	Topoisomerase I, CDK4/6	R/R neuroblastoma, medulloblastoma, HGG
Abemaciclib + Irinotecan + TMZ + Dinutuximab + GM-CSF	NCT04238819	1 and 2	CDK4/6, topoisomerase I, GD2	HGG
Nivolumab	NCT04323046	1	PD-1 receptor	Malignant, recurrent HGG
CD200 Activation Receptor Ligand (CD200AR-L) and Allogeneic tumor lysate vaccine + adjuvant re-irradiation	NCT06305910	1	CD200R1	DMG, H3K27M HGG
INCB7839	NCT04295759	1	ADAM	HGG, DIPG, DMG
SurVaxM + Montanide ISA 51	NCT04978727	1	IAP	HGG, DIPG
NKTR-214 + Nivolumab	NCT04730349	1 and 2	CD8T	HGG

NK = natural killer, RT = radiotherapy, DC = dendritic cell, TMZ = temozolomide, R/R = relapsed/refractory.

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
