# Peer review of "Emerging and Biological Concepts in Pediatric High-Grade Gliomas"

_cells, 2024, doi:10.3390/cells13171492_

Round 1

Reviewer 1 Report

Comments and Suggestions for Authors

Summary: the authors have written a comprehensive review on the current state of pediatric high grade gliomas.  The subject matter and scope are appropriate.  However, there are some issues with the content.  Very importantly, there have been many changes to the categorization of these tumors starting with WHO 2016 and then more recently 2021.  In some instances these seem to be confused.  Generally most sections would benefit from more specificity.  The immune section particularly lacks specificity.  There is a special responsibility inherent in review articles.Please see specific comments on content. 

Specific comments line by line:

30 – Historical terminology can still be relevant for describing these tumors but must be qualified as such.  Please clarify that Grade 3 AA and Grade 4 GBM are historical terminology now.

31 – What is Grade 21?  Assume this is a typo.  Please correct.  Also what is incidence referring to?  Global incidence?  Please specify. 

38 – why specific focus on supratentorial tumors here?

Figure 1 – placing the mutations in specific anatomic locations is confusing and can be misleading. 

66 – “H3K27M – mutant” is the WHO2016 terminology.  WHO 2021 uses “H3K27-altered”

69 – not just “H3K27M”

82 – the use of the term “more favorable prognosis” is misleading.  The survival is slightly longer but the prognosis is still dismal.   Please specify how these entities are different from a survival perspective.

89-90 incorrect reference notation

91 – please be as specific as possible when referencing survival

Section 2.4 IHG – Further discussion regarding historically dismal prognosis to chemo/rads but response to targeted therapies being evaluated and hopefully changing.

126 – what is meant by an “extended period”?  This is more typical of the diagnosis of MB, EPN and pLGG not pHGG which are more likely to be acute.

132 – the term “motor deficits” is too non specific.  Please specify the deficits and their relation to tumor locations.

138 – imaging is not a diagnostic modality here.  It is the way most tumors are initially identified during symptomatic work up but a surgical biopsy is usually necessary for diagnosis.

141- “stroke or other brain pathology” is the intent to discuss differential diagnosis here?  If so please elaborate.  Stroke and mass lesion work ups have important differences.

150 – please provide more information regarding the specific utility of different MRI sequences which is standard of care.  There is very little description here and then jumps to PET which is not a standard modality at most centers.

166 – CSF can be accessed by different approaches including EVD etc

167 – may be better to say that risk benefit should be carefully considered for each patient. 

187 – “H3K27-altered” WHO 2021 terminology should be used

208 – Is this specific to H3K27M mutations with p53 or any H3K27 alteration?

212 – specific to H3K27M or all H3K27 mutations?

219 – what does “signaling” refer to here?

229 – “improved OS” again specifics would be better when describing survival statistics

237 – “catastrophic telomere loss” - is chromothripsis implied here?  This description suggests a specific mechanism.  Please clarify.

239-245 – Danussi et al’s study is very well done.  However, this was regarding aHGG not pHGG.  Since astrocytic histology is the norm, how is this specific to ATRX mutant tumors?

264 – what about the role of targeted therapy in BRAF mutant pHGG?

267 – Germline NF-1 mutations behave completely differently from somatic ones.  While NF-1 patients typically develop LGG, it is not uncommon to find loss of NF-1 in HGG usually alongside other mutations.

294 – what is meant by “strong inflammatory profile” this is extremely nonspecific language.

295 – What is meant by “DHGs”?

298 – This is a very sweeping statement with only one reference.  What is meant by “macrophage function/inflammation/chemokines” vs “innate immunity”?  Macrophages are part of the innate immune system. 

305 – “lacking cytokines”  nonspecific terminology again.  Which cytokines? 

308 – “This suggests that, unlike adult brain tumors, pediatric brain tumors do not primarily rely on an immunosuppressive environment to evade the immune system”.  This is another sweeping statement that cannot be made bsed on the evidence provided. 

317 – while aHGGs may have more infiltrating immune cells than pHGGs it is misleading to say aHGGs have a “robust infiltrate”. 

318 – saying “immune infiltrate” does not correlate with survival is too nonspecific.  Is this regarding all CD45 cells or other populations.  Please be specific. 

Please carefully review the immunotherapy sections in “Emerging Therapies” Section 7 given the above.

Comments on the Quality of English Language

English Language: while English language is generally fine, there are quite a few instances where unusual phrases are repeated in close proximity.  This should be read and revised for clarity.  For example lines 27&29 the phrase “surpassing ALL” is used back to back.  89&91 – “16% of cortical tumors” used twice back to back.  200-202, “Various stress signals, including DNA damage, hypoxia, and chemotherapy, activated the p53 pathway, which in turn triggers different cellular responses such as cell cycle arrest, apoptosis, differentiation, DNA repair, and autophagy through intricate networks”  Followed by 204-206, “Dysfunction in TP53 in cancer cells can disrupt various cellular processes, including cell cycle arrest, apoptosis, differentiation, DNA repair, and autophagy, through intricate networks.”

There are also multiple places where the reference format is incorrect and some incorrect tense as well.  Please proof carefully.

Reviewer 2 Report

Comments and Suggestions for Authors

This is a very well written paper.  There are a few mistakes in the writing that can be easily corrected and are outlined below.  There are a few questions for the authors to consider.

The main issue is the limited discussion of new advances in the field of surgery and radiation in the treatment of Gliomas. In section 6.2, proton therapy should be discussed.

Recommended changes:

1. Introduction, line 29: "surpassing acute lymphoblastic leukemia is stated twice.  Remove one.

2. Line 89: Clarify "18"

3. Line 90: Clarify "5"

4. Line 171, 181: Reconsider the numbering system for the subtitles in the section.

5. Line 191: "However" is not needed here.

6. line 200: Consider the sentence structure.  Consider "activate" the p53 pathway "that" in turn trigger

7. Line 214: Do you mean "DMGs"?

8. Line 232: Do you mean "identified"

9. Line 263: Clarify "5"

10. Line 285: Should "Repotrectinib" be added (regardless of approval status)

11. Line 291: Remove "of"

12. Line 299: Consider changing "This" to "The"

13. Line 304: Clarify "104"

14. Line 307: Should this be "PD-1" instead of "PD-L1

15. Line 309: Clarify "107"

16. Line 318: Clarify "104"

17. Line 410: Should this be "effects"

18. Line 432-433: This is not a sentence as it stands.  Remove "such as"

19: Line 590: Should this be "Co"?

Comments on the Quality of English Language

The quality of the English is fine.  However, there are a few mistakes in the manuscript that will be easy to correct.

Round 2

Reviewer 2 Report

Comments and Suggestions for Authors

This is a good review of the current understanding of the biology of high grade gliomas.  There are mo significant issues.

A few very minor issues:

1. On line 28 ALL is a cause of death but not a significant cause of CNS mortality.  Just correct the English sentence structure

2. On line 91 change to "longer median"

3. On line 137 suggest changing "While" to "Although"

4. On line 138 suggest changing "which" to "that"